# Radiation-Variation Insensitive Coarse-to-Fine Image Registration for Infrared and Visible Remote Sensing Based on Zero-Shot Learning

Jiaqi Li [1,2], Guoling Bi [1], Xiaozhen Wang [1,2], Ting Nie [1] and Liang Huang [1,*]

[1] Changchun Institute of Optics, Fine Mechanics and Physics, Chinese Academy of Sciences, Changchun 130033, China; lijiaqi221@mails.ucas.ac.cn (J.L.); biguoling@ciomp.ac.cn (G.B.); wangxiaozhen22@mails.ucas.ac.cn (X.W.); nieting@ciomp.ac.cn (T.N.)

[2] University of Chinese Academy of Sciences, Beijing 100049, China

[*] Correspondence: huangliang@ciomp.ac.cn

**Abstract:** Infrared and visible remote sensing image registration is significant for utilizing remote sensing images to obtain scene information. However, it is difficult to establish a large number of correct matches due to the difficulty in obtaining similarity metrics due to the presence of radiation variation between heterogeneous sensors, which is caused by different imaging principles. In addition, the existence of sparse textures in infrared images as well as in some scenes and the small number of relevant trainable datasets also hinder the development of this field. Therefore, we combined data-driven and knowledge-driven methods to propose a Radiation-variation Insensitive, Zero-shot learning-based Registration (RIZER). First, RIZER, as a whole, adopts a detector-free coarse-to-fine registration framework, and the data-driven methods use a Transformer based on zero-shot learning. Next, the knowledge-driven methods are embodied in the coarse-level matches, where we adopt the strategy of seeking reliability by introducing the HNSW algorithm and employing a priori knowledge of local geometric soft constraints. Then, we simulate the matching strategy of the human eye to transform the matching problem into a model-fitting problem and employ a multi-constrained incremental matching approach. Finally, after fine-level coordinate fine tuning, we propose an outlier culling algorithm that only requires very few iterations. Meanwhile, we propose a multi-scene infrared and visible remote sensing image registration dataset. After testing, RIZER achieved a correct matching rate of 99.55% with an RMSE of 1.36 and had an advantage in the number of correct matches, as well as a good generalization ability for other multimodal images, achieving the best results when compared to some traditional and state-of-the-art multimodal registration algorithms.

**Keywords:** coarse-to-fine image registration; remote sensing; infrared; zero-shot learning

## 1. Introduction

Image registration refers to using different sensors, different times, or different locations to acquire two or more images with overlapping areas, unified to the same coordinate system so that the image homonymous points in the spatial location can be used to achieve optimal alignment [1]. Remote sensing technology is an emerging technology based on aerial photography that combines real-time sensing telemetry of the Earth's surface with resource management surveillance, and remote sensing image registration is the basis for utilizing remote sensing image information [2]. The registration of infrared and visible remote sensing images is an essential type of multi-sensor image registration. Infrared detectors detect the radiation information of objects in the scene [3], visible light detectors detect the reflection information of the objects, and the organic combination of the two can enhance the complementarity of scene information and reduce the uncertainty of understanding the scene. Therefore, it has received much attention in application areas

such as military intelligence acquisition [4], autonomous navigation [5], terminal guidance [6], target tracking [7], image fusion [8], change detection [9], and environmental monitoring [10].

At present, there are still more challenges in infrared and visible heterodyne remote sensing image registration, which can be summarized as follows:

1. The small number of features in the sparse texture region of multi-source remote sensing images leads to the problem of difficult matching. Visible images can better reflect the texture information in the scene with a clear hierarchy, while infrared images have less texture, similar structure repetitions, and fuzzy edges, which makes it difficult to distinguish the details in these images.

2. Heterogenous remote sensing images are difficult to acquire and screen, and the training of network models requires a large number of samples. Although there is a large amount of remote sensing image data available, datasets comprising real camera parameters, control points, or homography matrices as labels are scarce.

3. There are image grayscale distortions and image aberrations of different degrees, natures, and irregularity due to nonlinear spectral radiation variations during the acquisition of remote sensing images by different sensors. This radiation variation is a bottleneck problem limiting the development of multi-source remote sensing image matching techniques, and the seasonal and temporal phase differences also lead to large feature variations. As a result, the similarity between the corresponding locations of remote sensing images from different sources is weak, and it is difficult to effectively establish a large number of correct matches with the existing similarity metrics.

To address the problem of sparse texture, we propose a detector-free semi-dense registration algorithm. To address the problem of having few datasets available for supervised training on heterogeneous remote sensing imagery, we adopted the zero-shot learning method and propose a test dataset for manual labeling. For the problem of weak feature similarity due to radiation variation, we adopted the strategy of increasing reliability and transforming the matching problem into a model fitting problem to reduce the matching difficulty. Our main contributions are as follows:

1. RIZER, as a whole, employs a detector-free, end-to-end, coarse-to-fine registration framework, making the matching no longer dependent on texture and corner points. The innovative Transformer [11] architecture based on zero-shot learning in the field of infrared and visible remote sensing image registration improves the effectiveness of the pre-trained model, which makes the data-driven methods no longer limited by domain-specific datasets.

2. Knowledge-driven methods were adopted for the coarse-level matches, and the graph model-based K-nearest neighbor algorithm—Hierarchical Navigable Small World (HNSW) [12,13]—is introduced in the field of image registration for deep learning to efficiently and accurately obtain a wide range of correspondences. We also introduce the a priori knowledge between the matchpoints for local geometric soft constraints to build control point sets, which improves the interpretability and reliability of feature vector utilization and is not affected by radiation variation.

3. Simulating the strategy of first focusing on highly similar features before predicting the overall variation when the human eye is registered, the registration problem is transformed into a problem of fitting a transformation model through high-confidence control points, and multi-constrained incremental matching is used to filter between predicted matchpoints and establish a one-to-one matching relationship to achieve the overall insensitivity to radiation variations.

4. After fine-level coordinate fine-tuning, a simple but effective outlier rejection method that only requires extremely few iterations further improves the final matching results. A manually labeled test dataset of infrared and visible remote sensing images containing city, coast, mountain, desert, and aerial remote sensing images is proposed. Compared with classical and state-of-the-art registration algorithms, RIZER achieved competitive results. At the same time, it has an excellent generalization ability for

other multimodal remote-sensing images. Four ablation experiments were designed to demonstrate the effectiveness of the improved module.

## 2. Related Work

It is generally accepted that the registration algorithms for multimodal images are developed based on traditional images. Feature-based methods consist of four stages: feature detection, feature description, feature matching, and homography estimation [14]. Before the advent of deep learning, many handcrafted descriptor-based methods [15–17] were widely used for various registration tasks. The Scale-Invariant Feature Transform (SIFT) [18] algorithm is best known as a landmark work in the field of image registration, but it is strongly affected by nonlinear radiometric differences and is difficult to use directly for multimodal image registration. For multimodal remote sensing image registration [19–21], the PSO-SIFT [22] algorithm introduces a new definition of gradient based on SIFT, which improves the robustness of the descriptors to grayscale differences. Radiation-variation Insensitive Feature Transform (RIFT) [23] effectively resists nonlinear radial differences by utilizing phase congruency for feature point detection and a maximum index map for description. While most of these multimodal matching algorithms have relatively excellent resistance to nonlinear radiation differences, the handcrafted descriptors may not perform well when dealing with more complex matching tasks due to their design limitations based on existing knowledge.

In recent years, with the booming development of deep learning [24], its deep feature extraction and expression ability can make up for the defects of shallow feature instability and improve the robustness of the registration algorithm. Deep learning applied to feature-based registration methods is generally based on embedded modules, i.e., neural networks are used instead of traditional methods at a certain stage [25–27]. DescNet [28] performs a multilevel convolutional pooling operation on the input image block to obtain the deep features of the image block and finally outputs a 128-dimensional vector as a feature descriptor. ReDFeat [29] re-couples the independent constraints of detection and description of multi-constrained feature learning with a mutual weighting strategy and thus does not directly suppress the probability of detecting an ambiguous feature. SuperGlue [30] constructs a Graph Neural Network (GNN) for jointly finding correspondences and rejecting non-matchable points by treating the feature matching problem as solving a differentiable optimal transport problem. Modular networks cannot avoid the problem of error accumulation in traditional multi-stage registration, the feature detection methods used rely on edges, and corner points still have limitations, making it difficult to extract enough points of interest to obtain good descriptors for challenging edge blurring or texture-repetitive scenarios, leading to poor matching results.

With the application of the Transformer [11] model in the field of image processing, its chunking of the image, and extraction of the global features, a larger feeling field is realized. In order to solve the problems of modular networks, detector-free registration methods have been developed based on Transformer. It does not rely on feature point extraction and can directly perform end-to-end feature matching between image pairs. The most famous detector-free method is LoFTR [31], which constructs dense matching at the coarse-level and refines it at the fine-level to achieve sub-pixel matching. AspanFormer [32] adjusts the fine-level span of attention in an adaptive manner based on hierarchical attention. MatchFormer [33] interleaves self-attention for feature extraction and cross-attention for feature matching, improving the robustness of matching. For remote sensing image registration [34,35], SRTPN [36] adds a scale regression module and a rotation estimation module before the LoFTR module [31] to recover the geometric transformations. MIVI [37] employs a four-stage matching architecture based on LoFTR [31] for infrared and visible image registration, enabling the model to capture fine local feature details and remote dependencies. The cost of achieving global attention for the Transformer model [11] requires tens of thousands (or even more) of data samples to train a better model. At the same time,

it consumes a large amount of computational resources, requiring the use of large-scale Graphics Processing Unit (GPU) clusters for training.

To address the problem of difficult network training, transfer learning using pretrained network models has been shown to be very effective [38]. M2DT-Net [39] employs a fine-tuned, end-to-end network model to eliminate differences between two multimodal remote sensing images. For zero-shot learning, DFM [40] uses a pre-trained ResNet architecture [41] as a feature extractor and does not require any additional training to improve matching. However, the subsequent matching is established within the small receptive field of the initial matching. For some difficult matching situations, such as the registration of heterogeneous remote sensing images, matching points may be concentrated in only a few partial areas. To the best of our knowledge, there has been no systematic study of the effectiveness of utilizing the pre-trained model of the Transformer module [11] as a feature extractor in the field of heterologous remote sensing image registration, nor has there been a study to demonstrate this limitations. If this approach can fully utilize the global attention and semantic abstraction ability of the Transformer model, then the pre-trained alignment model using the visible scene can also achieve satisfactory performance in the task of heterogenous remote sensing image registration.

## 3. Methodology

### 3.1. Workflow of the Proposed Method

As shown in Figure 1, we propose an end-to-end algorithm for infrared and visible remote sensing image registration named RIZER. The algorithm is divided into three main stages. The first stage utilizes a pre-trained ResNet [41] combined with Transformer [11] for feature extraction. In the second stage, coarse-level matching broad correspondences are obtained using the Dual HNSW algorithm. Then, the local geometric soft constraints are utilized to filter the set of control points, after which model fitting is performed. Finally, an incremental matching method with multiple constraints is used. In the third stage, after combining the coarse-level matchpoint locations with the fine-level map, Transformer is then used for fine-level feature fusion, coordinating fine-tuning by calculating the expectation between the matchpoints, and finally, efficient outlier rejection is performed to obtain the final matching results. Among them, the second stage and the outlier rejection part in the third stage are our main focus, which will be described in detail below. Here, we briefly describe how the feature maps are generated in the first stage.

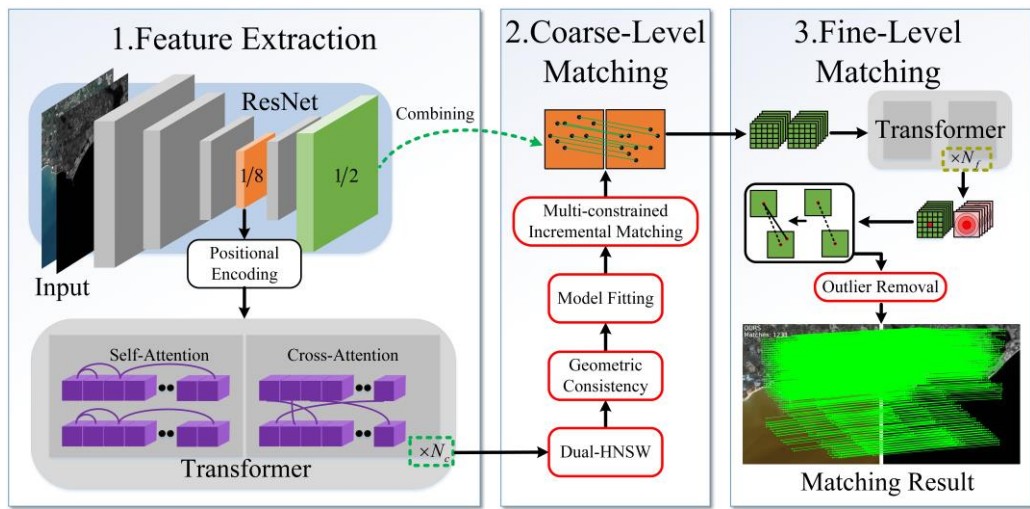

**Figure 1.** Flowchart of RIZER.

The infrared and visible remote sensing images are fed into the feature extraction network, and the standard convolutional structure of Feature Pyramid Networks (FPNs) is used to extract multilevel features from the two images. The initial features extracted by

the CNN have translational invariance. Thus, they can recognize similar patterns occurring at different locations in the image, and the FPN achieves multiscale feature fusion. The extracted 1/8-resolution feature maps are used to generate global context-dependent feature descriptors, and downsampling reduces the input length of the subsequent Transformer module. Then, 1/2-resolution feature maps are used to fine-tune the coordinates of the coarse matchpoints, which ensures that the final matchpoints have higher localization accuracy.

The 1/8-resolution feature maps are subjected to sine–cosine positional encoding, which makes the output features location-dependent and improves the feature representation of the model for sparse texture and repetitive regions. After that, the features are fed into the Transformer module for feature enhancement, which is repeated several times. The Transformer module consists of multiple alternating self-attention and cross-attention layers, where the self-attention layer allows each point to focus on all other points around it within an image to capture the correlation between them, while the cross-attention layer allows each point to focus on all the points in another image. By utilizing the excellent property of Transformer's global attention, the features of each point are fused with the contextual information of other related points, obtaining a richer representation, and coarse-level feature maps $F_c^A$ and $F_c^B$ are obtained.

Here, for the feature extraction network, we used the outdoor weights trained by the LoFTR [31] model. Due to the small labeled dataset of infrared and visible remote sensing images, we tried not to train the scene in a targeted way, but directly utilize the feature vectors obtained from the pre-trained weights for subsequent processing, i.e., zero-shot learning. The results show that such a strategy is efficient and feasible, and the feature maps are shown in the Results Section.

### 3.2. Dual HNSW

In order to achieve overall differentiability, current detector-free algorithms [31–33] commonly use the inner product between vectors to obtain the similarity matrix and then search for the maximum value to establish a match, e.g., the dual-softmax method used by LoFTR [31] to find the two-way maximum of the match probability. Although this strategy achieves differentiability in similarity measurement, it is too strict and does not fully utilize the relationship between the feature vectors, especially for infrared and visible multimodal images, which can filter out some of the correct matches or even establish incorrect matches. In this study, we adopted the strategy of increasing reliability, i.e., initially establishing one-to-many fuzzy correspondences and then establishing reliable matches through screening. Since we utilize pretrained weights, our subsequent matching strategy does not rely on differentiability. Therefore, we introduce a graph model-based K-nearest neighbor algorithm, HNSW [12], in the field of deep learning-based image registration. HNSW is an efficient approximate nearest neighbor search algorithm, which is not yet very widely used in the field of image registration. It accelerates the nearest-neighbor search process by constructing a hierarchical graph index structure and jumping between and within layers, which can significantly reduce the amount of distance computations that need to be performed.

The query index and template index are established, respectively, corresponding to the input template vector $F_c^A$ and the query vector $F_c^B$ for the search of K-nearest neighbors, to obtain the K-nearest neighbor set $D_k^A$ of each coarse-level point, as shown in Equation (1), as well as $D_k^B$, $k = 1, 2, 3$, where $d_k(i)$ stands for the K-nearest neighbor distance to the ith point in the template image, the default distance metric of HNSW is the square of P-paradigm, and $(\hat{x}_j, \hat{y}_j)$ is the corresponding point in the query image. Filtering is performed using the nearest neighbor and next nearest neighbor through the proportional approach in the SIFT [18] algorithm. A match is established if Equation (2) is satisfied, and here, *ratio* is set to 0.8. Unsatisfied or unused K-nearest neighbor relationships will be fully utilized in subsequent processing.

$$D_k^A(i) = \left\{ (\hat{x}_j, \hat{y}_j), d_k(i) \right\} \tag{1}$$

$$d_1(i)/d_2(i) \prec ratio \tag{2}$$

The threshold-filtered template vectors and the mutual nearest neighbors of the query vectors are used to establish preliminary matches, and the preliminary matched template point set $S_1^A$ and the query point set $S_1^B$ are obtained. This threshold-filtered bidirectional matching method can filter out most false matches and guarantee that the preliminary matches have a high in-points rate. The specific diagram is shown in Figure 2.

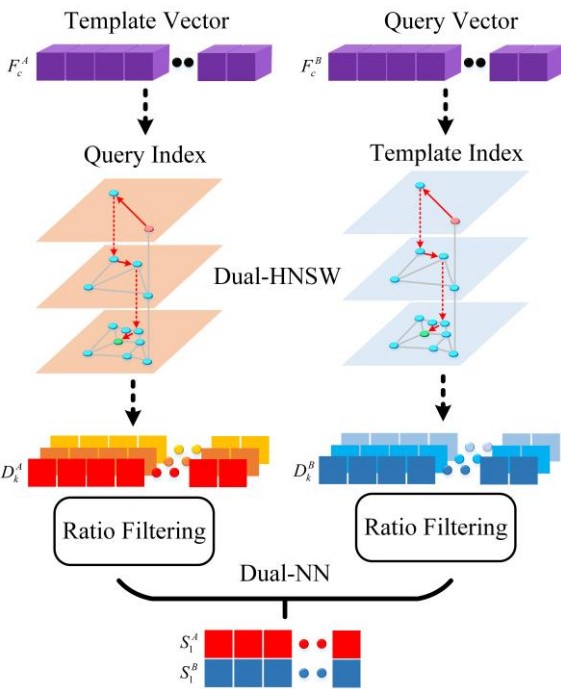

**Figure 2.** Schematic diagram of dual HNSW.

### 3.3. Local Geometric Soft Constraint

After the dual HNSW algorithm establishes the initial matching, some pseudo-correspondences may still exist due to the complexity of the heterogeneous remote sensing images, which will have a great impact on the fitting of the subsequent transformation model and even lead to the failure of the matching. We introduce the a priori information of geometric consistency [42] between matchpoints to establish constraints and then carry out another pseudo-correspondence rejection. Meanwhile, compared with the heat map approach, the introduction of a priori knowledge can improve the interpretability of the deep learning method, and the results it obtains are not irregular, but they all satisfy the geometric consistency, which makes the obtained registration results more reliable. When the image does not contain severe affine or projective transformations, the spatial structure between correct matchpoints is similar. Based on this feature, the mismatched matchpoints can be eliminated by calculating the Euclidean distance and direction angle between their neighboring matchpoints.

The difficulty in matching between heterologous remote sensing images in infrared and visible is mainly caused by different imaging mechanisms, considerable nonlinear spectral radiation variations, and obvious differences in the grayscale gradient, whereas geometric structural attributes are less affected by radiation differences. Heterologous remote sensing images captured by satellites usually do not have a large perspective deflection in a small area, and the viewing angle of satellite remote sensing images is relatively high. The ground can be regarded as an approximate plane, and the depth of the scene is relatively small, which makes the remote sensing images a two-dimensional planar projection, which satisfies the condition of spatial structure similarity.

The core idea of geometric consistency is that for all correct matchpoint pairs, the distance ratios of any two matchpoint pairs should be equal or approximately equal, and the distance ratios should also be approximately equal to the true scale ratios of the two

images and the angles formed by any three matchpoint pairs should be approximately equal. So, the distance ratios of all correct matchpoint pairs can form a class centered on the true scale ratio. The matchpoint pairs that are farther away from the center of the class are the false matches, and the angles between the matchpoint pairs should also satisfy the corresponding constraints.

In this study, we build on the preliminary matching point sets $S_1^A$ and $S_1^B$ for culling in order to improve the efficiency of the algorithm; for each matching point pair, any five-point pairs among the remaining matching point pairs are selected to establish two corresponding local random matrices, where $R^A$ is the local random matrix of the template image, as shown in Equation (3), and $R^B$ is the local random matrix of the query image. $A_{u,0}$ stands for the coordinates $(x_{u,0}, y_{u,0})$ of the uth point as the center of the point; $A_{u,v}, v = 1, 2, \cdots, 5$, stands for the coordinates $(x_{u,v}, y_{u,v})$ of the randomly selected points among the remaining matchpoints. $u = 1, 2, \cdots, n$, $n$ represents the number of matchpoints established by the preliminary matching.

$$R^A = \begin{bmatrix} A_{1,0} & A_{1,1} & A_{1,2} & A_{1,3} & A_{1,4} & A_{1,5} \\ A_{2,0} & A_{2,1} & A_{2,2} & A_{2,3} & A_{2,4} & A_{2,5} \\ \vdots & \vdots & \vdots & \vdots & \vdots & \vdots \\ A_{n,0} & A_{n,1} & A_{n,2} & A_{n,3} & A_{n,4} & A_{n,5} \end{bmatrix} \tag{3}$$

The Euclidean distance is used to calculate the distance $L_{u,v}$ between $A_{u,v}$ and $A_{u,0}$, as shown in Equation (4), as well as to calculate $\hat{L}_{u,v}$ through the matrix $R^B$ to form the length matrices $L^A$ and $L^B$.

$$L_{u,v} = \sqrt{(x_{u,v} - x_{u,0})^2 + (y_{u,v} - y_{u,0})^2} \tag{4}$$

$$\theta_{u,v} = \begin{cases} \arccos \frac{(x_{u,v} - x_{u,0}) \times (x_{u,v+1} - x_{u,0}) + (y_{u,v} - y_{u,0}) \times (y_{u,v+1} - y_{u,0})}{L_{u,v} \times L_{u,v+1}}, v = 1, 2, 3, 4 \\ \arccos \frac{(x_{u,5} - x_{u,0}) \times (x_{u,1} - x_{u,0}) + (y_{u,5} - y_{u,0}) \times (y_{u,1} - y_{u,0})}{L_{u,5} \times L_{u,1}} \end{cases} \tag{5}$$

To further improve the computational efficiency, only the angle between the current edge and the next edge is computed, while the fifth edge forms an angle with the first edge, and the computation yields $\theta_{u,v}$, as shown in Equation (5), which composes the angle matrix $\theta^A$, and $\hat{\theta}_{u,v}$ is computed by query image and composes the angle matrix $\theta^B$. A schematic diagram of the established double feature constraints of the length and the angle is shown in Figure 3.

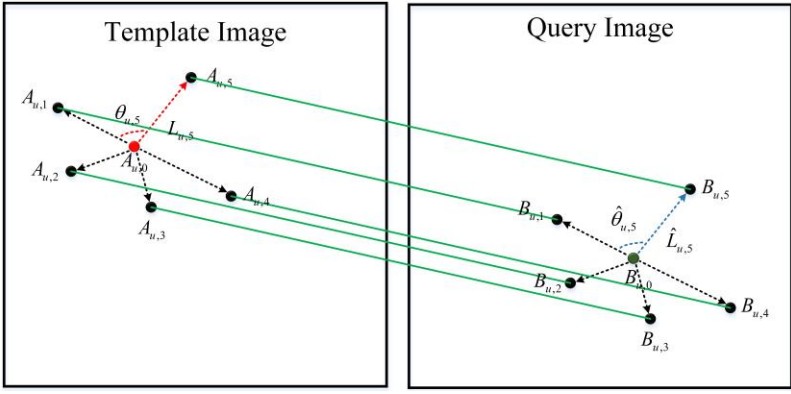

**Figure 3.** Schematic diagram of local geometric consistency.

After calculating the length matrices $L^A$ and $L^B$ and the angle matrices $\theta^A$ and $\theta^B$, it is necessary to judge whether the ratio between them satisfies the a priori information of geometric consistency. In this study, we adopted the method of soft constraints, i.e., instead of requiring each distance and pinch angle to be within the threshold range, we take their overall error mean as a soft constraint, which reduces the number of judgments and avoids

making the algorithm too strict while also reducing the influence of a small number of outlier points, and this excludes matchpoints that do not satisfy the soft constraints as outlier points.

Specifically, the mean value of the length ratio $\bar{r}$ is first calculated, as shown in Equation (6), and then the length error $\Delta_{u,v}$ is calculated, as shown in Equation (7). For the angle, the ratio should be approximately equal to 1. The angle error $\varepsilon_{u,v}$ is calculated using Equation (8). Finally, the mean length error and the mean angle error are calculated for each pair of matchpoints, and if they are not simultaneously within the allowable error range, then the pair of pseudo-matches is eliminated, and the remaining pairs of matchpoints that satisfy the conditions of Equation (9) are summarized as the control point pairs $A_k(x_k, y_k)$ and $B_k(\hat{x}_k, \hat{y}_k)$, $k = 1, 2, \cdots, m$, and $m$ is the number of pairs of matchpoints of the control point set, which constitute the control point set $P_1^A$, as well as $P_1^B$, where the threshold $\tau$ is set to 0.1.

$$\bar{r} = \left[ \sum_{u=1}^{n} \sum_{v=1}^{5} \left( L_{u,v} / \hat{L}_{u,v} \right) \right] / (n \times 5) \tag{6}$$

$$\Delta_{u,v} = \left| \left( L_{u,v} / \hat{L}_{u,v} \right) / \bar{r} - 1 \right| \tag{7}$$

$$\varepsilon_{u,v} = \left| \left( \theta_{u,v} / \hat{\theta}_{u,v} \right) - 1 \right| \tag{8}$$

$$\left[ \left( \sum_{v=1}^{5} \Delta_{u,v} \right) / 5 < \tau \right] \wedge \left[ \left( \sum_{v=1}^{5} \varepsilon_{u,v} \right) / 5 < \tau \right] \tag{9}$$

*3.4. Least Squares Fitting Transform Model*

In recent years, many approaches have tended to formulate the registration problem as an optimization problem [43,44]. The prevailing principle is to optimize the transformation model to obtain the best possible registration results, which can be measured by minimizing an objective function that measures the registration's accuracy. The method of fitting a change model can avoid the direct use of complex feature vectors to align between the full matchpoint sets and converting the matching problem into a model fitting problem greatly reduces the difficulty of matching and avoids complete blind matching. At the same time, it is also in line with the process used by the human eye for registration; first, it finds the general rule of change among images through the significant matchpoint sets and then makes corresponding predictions and judgments one by one.

Fitting the change model is commonly used to estimate the affine transform model coefficients based on Random Sample Consensus (RANSAC) [45] and its improved algorithms, but the coarse-level map also destroys the original geometrical structure of the image based on the structure of the single-response matrix, which may lead to inaccurate estimation of the single-response matrix if the structure is changed. And, based on high-order nonlinear change models, such as the Thin-Plate Spline (TPS) [44] model, if the complexity of the algorithm is too high, it will significantly increase the running time. In addition, because the coarse-level map extracts semi-dense local feature points at certain intervals in the image, which are uniformly distributed in the whole image with strong regularity, the spatial transformation model is essentially a coordinate mapping function, and because the coarse-level map has a low resolution, it can greatly reduce the nonlinear variation between matchpoints, so the coordinates of matchpoints of the coarse-level map can be approximated as a linear relationship.

For a pair of control points $A_k(x_k, y_k)$ and $B_k(\hat{x}_k, \hat{y}_k)$, a polynomial fit is used, as shown in Equation (10).

$$\begin{cases} \hat{x}_k = ax_k + by_k + cx_ky_k + d \\ \hat{y}_k = ex_k + fy_k + gx_ky_k + h \end{cases} \tag{10}$$

Which reduces to a least squares matrix in the form of:

$$\begin{pmatrix} \hat{x}_k & \hat{y}_k \end{pmatrix} = \begin{pmatrix} x_k & y_k & x_k y_k & 1 \end{pmatrix} \begin{pmatrix} a & e \\ b & f \\ c & g \\ d & h \end{pmatrix} \tag{11}$$

Substituting the control point sets into the above equation:

$$\begin{pmatrix} \hat{x}_1 & \hat{y}_1 \\ \hat{x}_2 & \hat{y}_2 \\ \vdots & \vdots \\ \hat{x}_m & \hat{y}_m \end{pmatrix} = \begin{pmatrix} x_1 & y_1 & x_1 y_1 & 1 \\ x_2 & y_2 & x_2 y_2 & 1 \\ \vdots & \vdots & \vdots & \vdots \\ x_m & y_m & x_m y_m & 1 \end{pmatrix} \begin{pmatrix} a & e \\ b & f \\ c & g \\ d & h \end{pmatrix} \text{ i.e., } P_1^B = P_1^A W \tag{12}$$

Then, the transformed model matrix is solved:

$$W = P_1^{AT} P_1^B \left( P_1^{AT} P_1^A \right)^{-1} \tag{13}$$

*3.5. Multi-Constraint Incremental Matching*

Most of the matching algorithms use a single constraint for a one-size-fits-all judgment of interior and exterior points, which is difficult to adapt to the complexity of registration between remote sensing images from different sources. A single constraint will inevitably produce a misjudgment, resulting in a large number of interior points being eliminated, and the small number of matchpoints obtained does not guarantee a high interior point rate. To avoid this situation, single-constraint algorithms will commonly use an iterative elimination of outer points, which again leads to the inefficiency of the algorithm. In this study, we adopted a multi-constraint incremental matching approach, which combines multiple in-point prediction and out-point rejection methods to provide more potential one-to-one matches while guaranteeing a high in-point rate with high algorithmic efficiency.

First, based on the transform model constraints, $S_2^A$ and $S_2^B$ remain after removing the preliminary matches $S_1^A$ and $S_1^B$. Using $S_2^A$ and the solved transform model matrix $W$, the prediction is performed in the query image with rounding as well as the elimination of the obvious erroneous points outside the image to obtain $\widetilde{S}_2^B$ and the corresponding $\widetilde{S}_2^A$, to ensure that each point in $\widetilde{S}_2^B$ falls within $S_2^B$. After that, the mutual three-nearest-neighbor constraints are performed using the three-nearest-neighbor matrices previously obtained by the dual HNSW algorithm. It is too harsh to consider only the nearest-neighbor relationships in the heterologous remote sensing images and tends to exclude a large number of correct correspondences. Using mutual three-nearest-neighbor constraints can efficiently obtain most of the correct correspondences, make full use of the data obtained from the establishment of the dual index in the previous section, and judging point by point so that only when the point $(\hat{x}_j, \hat{y}_j)$ of $\widetilde{S}_2^B$ and the corresponding point $(x_i, y_i)$ of $\widetilde{S}_2^A$ are mutual three-nearest-neighbors to each other, are they saved as inner points, and we obtain the mutual three-nearest-neighbor matchpoint sets $P_2^A$ and $P_2^B$. The rest of the pairs of points that cannot satisfy the conditions at the same time are saved into the matrices $S_3^A$ and $S_3^B$. Finally, double feature constraints are then applied to $S_3^A$ and $S_3^B$ using length and angle, although the two point sets do not satisfy the nearest-neighbor constraints, it is possible that the obtained feature vectors and the HNSW algorithm cannot find the correspondence between them, and the spatial structural similarity between the matchpoints can be used as an additional source of matching. The specific implementation is similar to the above, but the first column of the random matrix consists of the points in $S_3^A$ and $S_3^B$, the last five columns are randomly selected pairs of points in the control point sets $P_1^A$ and $P_1^B$, and finally, the spatial structural similarity matchpoint sets $P_3^A$ and $P_3^B$ are obtained. Eventually, the set of all coarse-level matchpoints obtained, $P_c^A$ and $P_c^B$, consists of the control point sets $P_1^A$ and $P_1^B$, the mutual three-nearest-neighbor matchpoint sets $P_2^A$ and $P_2^B$, and the

spatial structural similarity matchpoint sets $P_3^A$ and $P_3^B$, and is composed of three parts, as shown in Equation (14).

$$P_c^A = P_1^A \cup P_2^A \cup P_3^A \tag{14}$$

*3.6. Fine-Level Matching*

After establishing the coarse-level matches and finding the approximate correspondences in the local area, fine-level matching is needed to fine-tune the coordinate positions. Here, LoFTR's [31] coordinate refinement adjustment strategy and pre-training weights are used. For each coarse-level match, its position is mapped in the fine-level map, and then a pair of local windows of size $5 \times 5$ are cropped out, which is then inputted into the Transformer [11] module to perform $N_f$ times the feature fusion to obtain the fine-level feature maps $F_f^A$ and $F_f^B$. After that, the center vector of the template local window is similar to all the vectors of the query local window to obtain the heat map, and by calculating the expectation of the probability distribution, the final position with sub-pixel accuracy can be obtained in the query window.

However, the LoFTR [31] algorithm does not incorporate the outlier rejection method. Although the overall correct matching rate of the algorithm is high, a small number of false matches will have a large impact on the subsequent homography estimation and various scenarios. Iteratively Re-weighted Least-Squares (IRSL) [46] and RANSAC [45] are commonly used in the traditional methods for false matchpoint rejection, and both are robust estimation methods based on the iterative strategy, which needs to be optimized through the process of repeated optimization to suppress outlier points. Traditionally, iterative outlier removal conducted across all matched points generally requires hundreds or thousands of iterations, leading to low computational efficiency. Additionally, it struggles with robustness and attaining high correct matching rates when encountering numerous outliers. In this paper, we present a simple yet effective improvement to the RANSAC algorithm, building upon the multi-stage matching approach described previously. We invoke OpenCV's findHomography function using RANSAC with default parameters. Instead of sampling across all matches, we leverage the high confidence set of control points with refined coordinates to estimate the homography matrix $H$. Subsequently, the template point $(x_i, y_i)$ is mapped to obtain the ideal corresponding point, which is categorized as an interior point if it is kept within the distance of threshold $T = 2$ from the query point $(\hat{x}_i, \hat{y}_i)$, as shown in Equation (15). Therefore, the proposed algorithm enables efficient outlier removal with an extremely low iteration count. Simultaneously, it resolves robustness issues due to the high correct matching rate attained by the control point set, ultimately obtaining the fine-level matching point sets $P_f^A$ and $P_f^B$.

$$\| H \cdot (x_i, y_i) - (\hat{x}_i, \hat{y}_i) \|_2 \leq T \tag{15}$$

**4. Experiments**

*4.1. Datasets*

To test the performance of the algorithm in sparsely textured regions, such as mountains and coasts, and similarly structured repetitive regions, such as deserts and cities, we created our own small-scale test dataset containing 40 pairs of images acquired by the Sentinel 2 satellite. In order to test the robustness of the algorithm against temporal phase differences, we acquired infrared and visible remote sensing images at different times with an image resolution of $512 \times 512$. Detailed information is shown in Table 1 and Figure 4a–d. The visible images were fused with the 2–4 bands of Sentinel 2, with a band range of 490 nm–665 nm and a spatial resolution of 10 m. The infrared images were fused with the 11–12 bands of Sentinel 2, with a band range of 1610 nm–2190 nm and a spatial resolution of 20 m. The images were resampled to 10 m. In order to simulate the initially acquired remote sensing images as much as possible, we adopted a 4-point parameterization [47] with a coordinate shift range of $[-150, 150]$, which can realize a certain degree of non-rigid

transformation, and each pair of images was manually labeled with ten pairs of sampling points to fit the homography matrix labels.

**Table 1.** Details of the Sentinel 2 dataset.

| Category | VIS Date | NIR Date | Position |
|---|---|---|---|
| Desert | 28 November 2015 | 28 December 2015 | Bayingolin, Xinjiang, China |
| Coast | 8 January 2016 | 7 February 2016 | Shantou, Guangzhou, China |
| City | 25 December 2015 | 7 August 2015 | Beijing, China |
| Mountain | 19 January 2016 | 18 April 2016 | Garz, Sichuan, China |

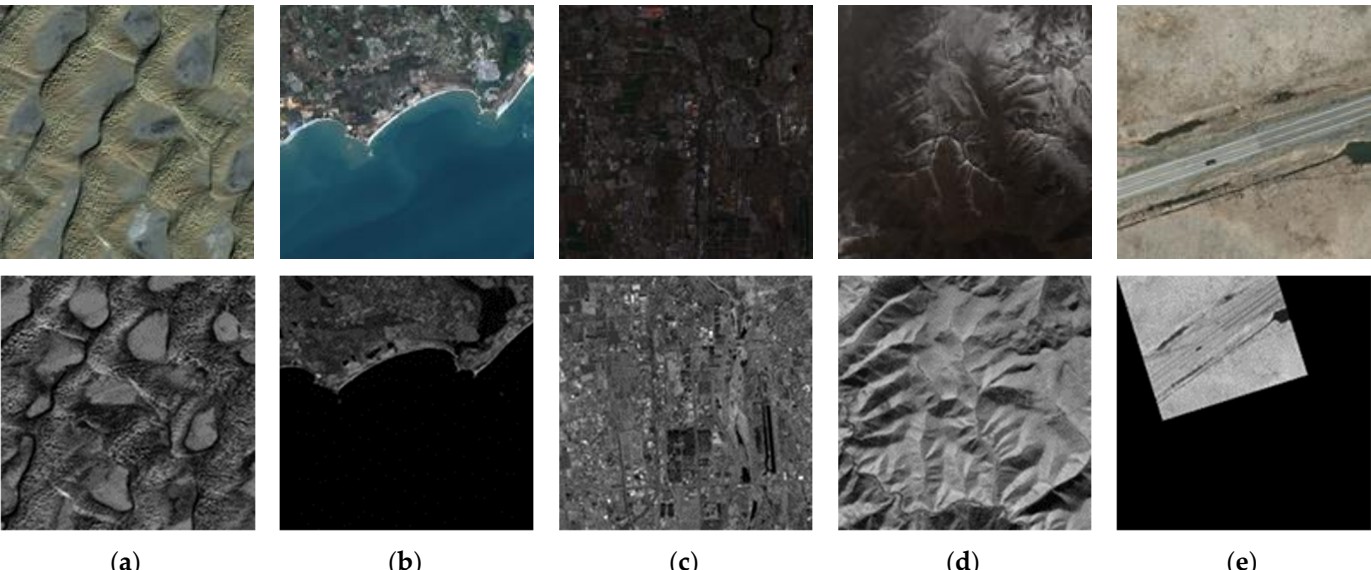

| (a) | (b) | (c) | (d) | (e) |

**Figure 4.** Infrared and visible remote sensing image dataset. (**a**) Desert, (**b**) coast, (**c**) city, (**d**) mountain, and (**e**) aerial images.

We conducted further tests on a publicly available aerial remote sensing dataset [48], which was selected as a data source for high-resolution aerial images captured by Utah AGRC in the spring of 2012, which consist of orthorectified projections containing visible and near-infrared channels with a spatial resolution of 12.5 cm, and an adopted image resolution of 512 × 512. A total of 50 pairs of images were randomly selected for testing. In order to test the robustness of the algorithm to rotation and scale variations, we randomly generated the transformations with translation factor $dx, dy \in [-0.5, 0.5]$, rotation angle $\alpha \in [-30°, -25°] \cap [25°, 30°]$, and scale factor $s \in [0.5, 1]$ for the infrared images, as shown in Figure 4e, and the corresponding labels of the homography matrices were generated.

In order to further validate the performance of the pre-trained model and the generalization ability of RIZER, the open-source multimodal remote sensing dataset was tested [49]. The details of the test dataset are shown in Table 2 and Figure 5, and the modalities of the test data include visible light, SAR, luminous remote sensing images, grid maps, and depth maps. Visible-Visible indicates the combination of two visible light images. Visible -SAR refers to visible light and SAR image pair. Day-Night denotes day and night image pair. Map-Visible represents map and visible light image pair. Visible-LiDAR shows the combination of visible light image and LiDAR data.

**Table 2.** Multimodal dataset details.

| Category | Template Image Sensor | Query Image Sensor | Size | Image Characteristic |
| --- | --- | --- | --- | --- |
| Visible–Visible | Google Earth | Google Earth | $600 \times 600$ | Different times |
| Visible–SAR | ZY-3 PAN optical | CF-3 SL SAR | $1000 \times 1000$ | Different bands |
| Day–Night | Optical | SNPP/VIIS | $1000 \times 1000$ | Day–night |
| Map–Visible | Open Street Map | Google Earth | $512 \times 512$ | Different models |
| Visible–LiDAR | WorldView-2 optical | LiDAR depth | $512 \times 512$ | Different models |

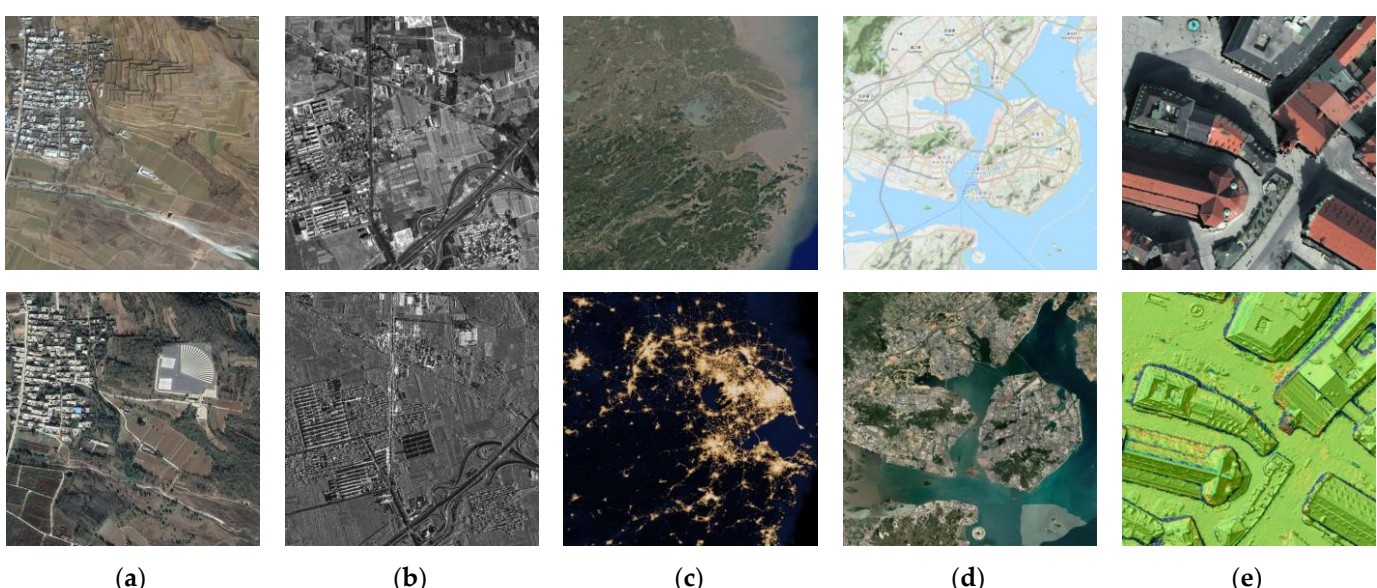

(**a**)  (**b**)  (**c**)  (**d**)  (**e**)

**Figure 5.** Multimodal remote sensing image dataset. (**a**) Visible–Visible; (**b**) Visible–SAR; (**c**) Day–Night; (**d**) Map–Visible; and (**e**) Visible–LiDAR.

## 4.2. Baseline and Metrics

Using three datasets, we compared the designed algorithm with five excellent algorithms, including some classical and the latest algorithms. The classic algorithms include SIFT [18] and the multimodal matching algorithm RIFT [23]. The latest algorithms include LoFTR [31], which consists of two coarse matching strategies, the dual-softmax method adopted by the original text, called LoFTR-DS, and the optimal transport method adopted by SuperGlue [30], called LoFTR-OT, and we tested them against the outdoor data trained with each of the two algorithms. In addition, we compared our algorithm to the latest ReDFeat [29] multimodal algorithm, which uses the corresponding VIS-NIR weights, and the multimodal module uses the best result among the weights. The test results in this study were obtained using a computer with a 4-core, 8-thread Intel i3-12100F@3.3/4.3 GHz CPU and an 8GB RTX4060 GPU.

For qualitative evaluation, we used the matchpoint connectivity diagram and registration checkerboard diagram. For quantitative evaluation, we used four indicators: Number of Correct Matches (NCM) [49], Success Rate (SR), Root Mean Square Error (RMSE) [50], and Running Time (RT). The formula for NCM is the same as Equation (15), and H represents the ground truth homography matrix. The threshold $T = 3\sqrt{2}$, when it is satisfied, is summarized as NCM and is represented as a green line in the matchpoint connectivity diagram; when it is unsatisfied, it is a wrong match and is represented as a red line. SR represents the percentage of the number of correct matches to the number of all matchpoints. RMSE represents the positioning accuracy of the correct matches and is computed using

Equation (16). $(x_i, y_i)$ and $(\hat{x}_i, \hat{y}_i)$ are the correct matches in the template image and query image, respectively. RT represents the total running time of the algorithm.

$$RMSE = \frac{1}{NCM} \sum_{i}^{NCM} \| H \cdot (x_i, y_i) - (\hat{x}_i, \hat{y}_i) \|_2 \qquad (16)$$

The larger the NCM and SR values are, the better the registration and the smaller the RMSE and RT values are, the better.

### 4.3. Registration Experiment for Infrared and Visible Remote Sensing Images

We used Principal Component Analysis (PCA) to downscale the high-dimensional features obtained from the coarse-level map and the downscaled results were visualized as RGB images, zooming in to view the detailed information to make an intuitive visual comparison of the two high-dimensional features. As shown in Figure 6, for the desert region with similar structural repetitions and the coastal region with sparse textures, the features extracted using the coarse-level map after zero-shot learning correspond to the same regions well. For example, the desert region can be roughly visualized into three parts: the lower left region, the diagonal region, and the upper right region, while the coastal region can clearly distinguish the land from the sea, and for the local similarity details, the visualization of the features in different colors indicates that the corresponding features are unique. The visualization results for the feature map show that the use of zero-shot learning has a certain degree of feasibility, the pre-training weights can extract enough distinguishable features, and, more importantly, we can see how to subsequently make full use of these features to establish registration (which is our main improvement).

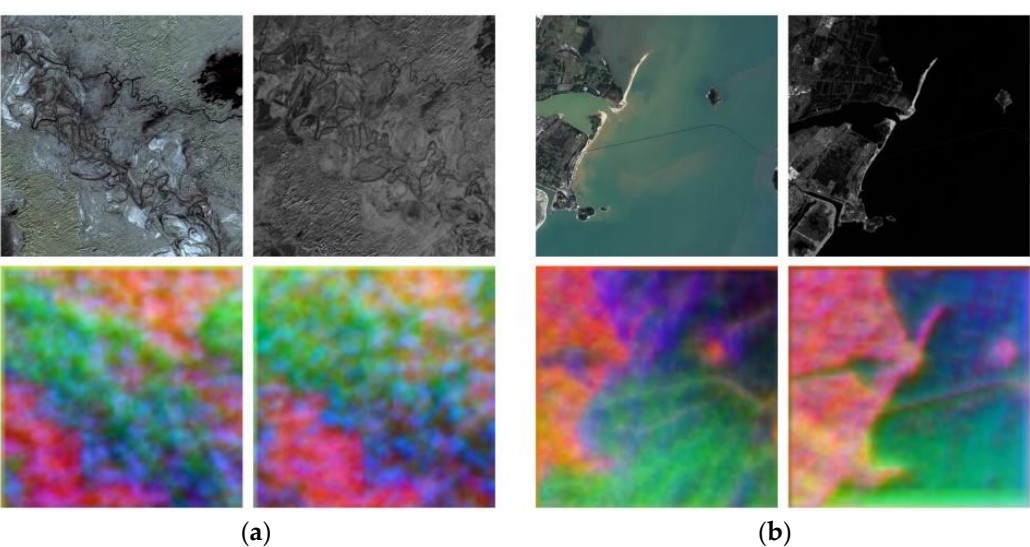

(**a**)             (**b**)

**Figure 6.** Feature visualization of infrared and visible remote sensing images. Visualization of infrared and visible remote sensing images and corresponding feature maps in (**a**) desert regions and (**b**) coastal regions.

We compared the performance of RIZER with that of five algorithms using the infrared and visible remote sensing image dataset above. Some qualitative matchpoint connectivity diagrams can be found in the Supplementary Materials Figure S1. Figure 7 shows the quantitative metrics of the algorithms in each scene. Figure 7a shows that RIZER had a slightly lower NCM than the ReDFeat algorithm in the coastal scene, while it achieved the highest number of matchpoint pairs in the rest of the scenarios, which is indistinguishable from our incremental matching strategy that is able to fully exploit the potential one-to-one matches. Between different scenes, for the desert and city scenes with more textures and repetitions, the number of matchpoint pairs of each algorithm was generally higher than that of the coastal and mountainous scenes with sparse textures, while the aerial images,

due to the presence of pure black regions, have the highest number of matchpoints between the two scenarios, which is in line with our baseline expectations. RIZER outperformed the other five algorithms in all scenarios in the SR and RMSE metrics, demonstrating the effectiveness of our multi-constraint strategy, the accuracy of the model fitting, and the necessity of the outlier rejection algorithm. The higher correct matching rate of coarse-level matching can indirectly improve the accuracy of the fine-tuned coordinates by reducing the self-attention and cross-attention between the matchpoints with the wrong matches in the fine-level stage. Figure 7d shows that the RT of RIZER was slightly higher compared to the LoFTR algorithm before improvement, mainly because the improved coarse-level matching part of the local consistency soft constraint is slightly cumbersome, while the ReDFeat algorithm has a great advantage in real time.

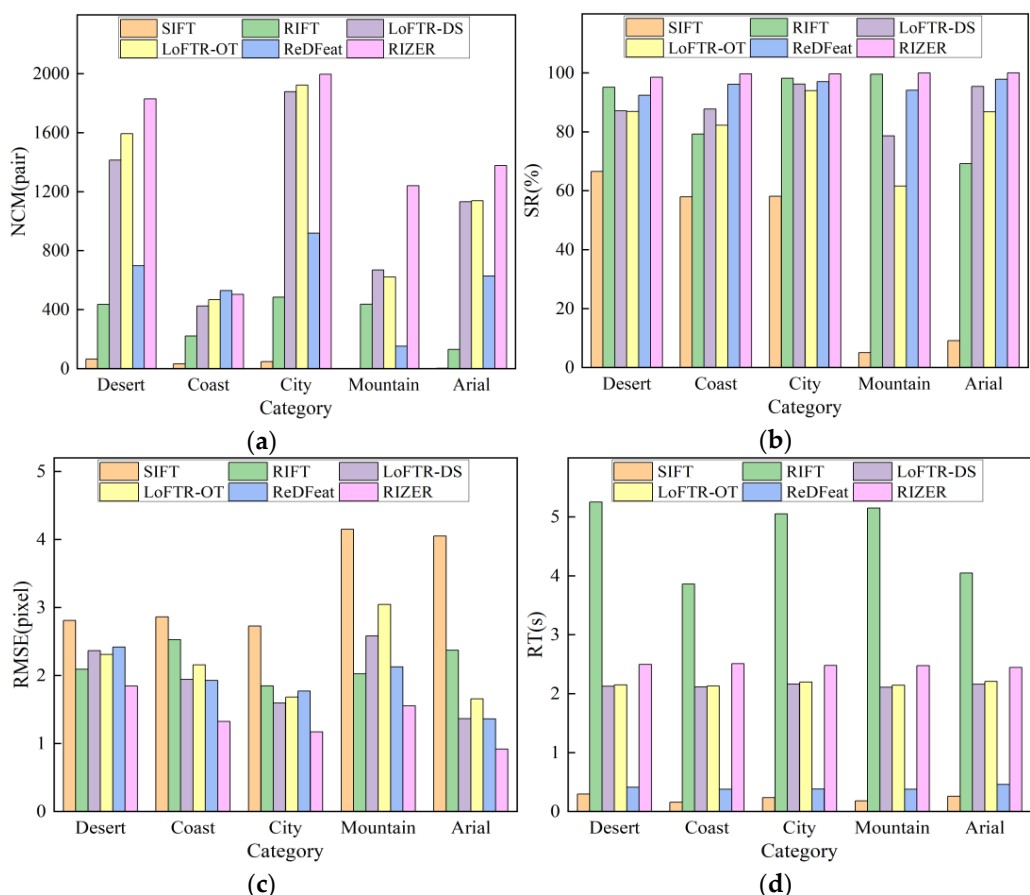

**Figure 7.** Indicator evaluation of the results of different algorithms for the registration of infrared and visible remote sensing images. (**a**) NCM; (**b**) SR; (**c**) RMSE; and (**d**) RT.

Table 3 shows the mean indicator values of the algorithms in each scene. It can be concluded that, for the infrared and visible remote sensing image registration in this scenario, RIZER improved the NCM by 25.9% compared to the pre-improvement LoFTR algorithm at the expense of 16.4% of the RT and achieved an SR of 99.5% with an RMSE of 1.36 pixels, which is comparable to the results of the state-of-the-art multimodal registration algorithms. Figure 8 shows the splicing effect of the tessellated map and the local zoomed-in image after the registration. This shows that the homography matrix fitted by RIZER had almost no difference compared with the real transformation and achieved an excellent registration effect.

**Table 3.** Mean indicator values of the results of different algorithms for the registration of infrared and visible remote sensing images.

| Algorithm | NCM (Pair) | SR (%) | RMSE (Pixel) | RT (s) |
|-----------|-----------|--------|--------------|--------|
| SIFT | 29.61 | 39.34 | 3.32 | 0.23 |
| RIFT | 341.83 | 88.25 | 2.17 | 4.67 |
| LoFTR-DS | 1103.71 | 89.00 | 1.97 | 2.13 |
| LoFTR-OT | 1148.92 | 82.29 | 2.17 | 2.16 |
| ReDFeat | 585.06 | 95.49 | 1.92 | 0.40 |
| RIZER | 1389.44 | 99.55 | 1.36 | 2.48 |

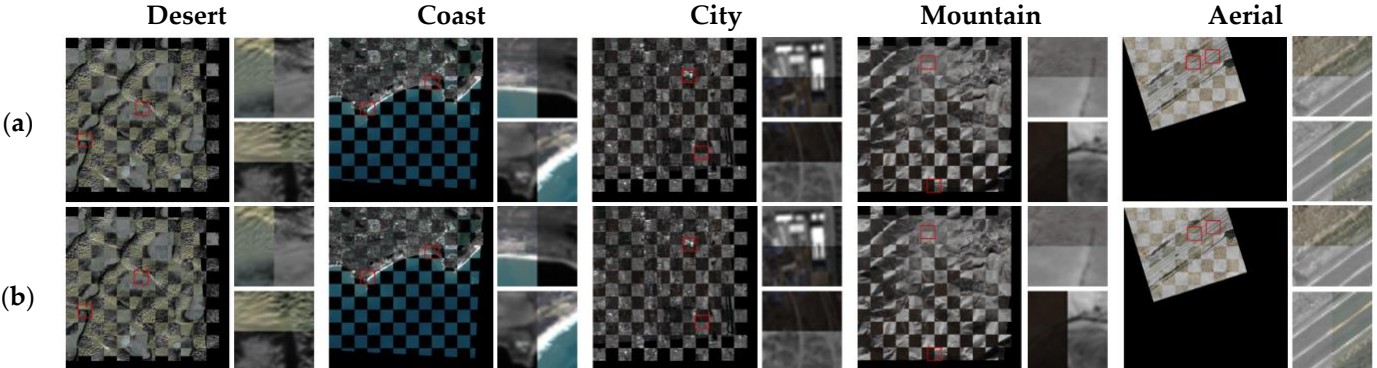

**Figure 8.** Registration checkerboard diagram of infrared and visible remote sensing images. (**a**) The real transformation; (**b**) RIZER.

*4.4. Registration Experiment for Multimodal Remote Sensing Images*

We compared the performance of RIZER with that of the five algorithms using the multimodal remote sensing image dataset. RIZER still had a strong generalization ability in multimodal scenes. Some qualitative matchpoint connectivity diagrams can be found in the Supplementary Materials Figure S2. Figure 9 shows the quantitative metrics of the algorithms for each scene. Figure 9a shows that the NCM of RIZER was slightly lower than that of the RIFT algorithm in the map–optical scene, while it achieved the highest number of matchpoint pairs in all the remaining scenes. Figure 9b shows that RIZER had a lower SR in the map–optical scene and Optical–LiDAR scene, while in the remaining three scenes, it achieved the highest correct matching rate compared to the other five algorithms, indicating that it is better suited for infrared and visible remote sensing image registration. Figure 9c shows the RMSE, which yields similar conclusions to that with the SR, with multimodal matching not achieving a superior performance, although it was competitive. Figure 9d shows that RIZER had a running time of about 2.7 s on 600 × 600 resolution images and about 3.3 s on 1000 × 1000 resolution images, while the RIFT algorithm had the disadvantage of a longer running time.

Table 4 shows the mean indicator values of the algorithms in each scene. It can be concluded that RIZER has some generalization ability in multimodal scenes and still achieved competitive results compared to the state-of-the-art multimodal registration algorithms. RIZER still had a very large advantage in the NCM and RMSE metrics but compared with the infrared and visible remote sensing image scenes, the correct matchpoint rate decreased by 5.6%, and the RMSE increased by 63.2%, so it is more suitable to be used in infrared and visible remote sensing image scenes. Figure 10 shows the splicing effect of the tessellated map after the alignment and the local zoomed-in image, which shows that the overall difference between the homography matrices fitted by RIZER was relatively small, but there were small splicing seams in some local places, such as the right side of the upper local map of Optical–Optical and the right side of the lower local map of Optical–LiDAR.

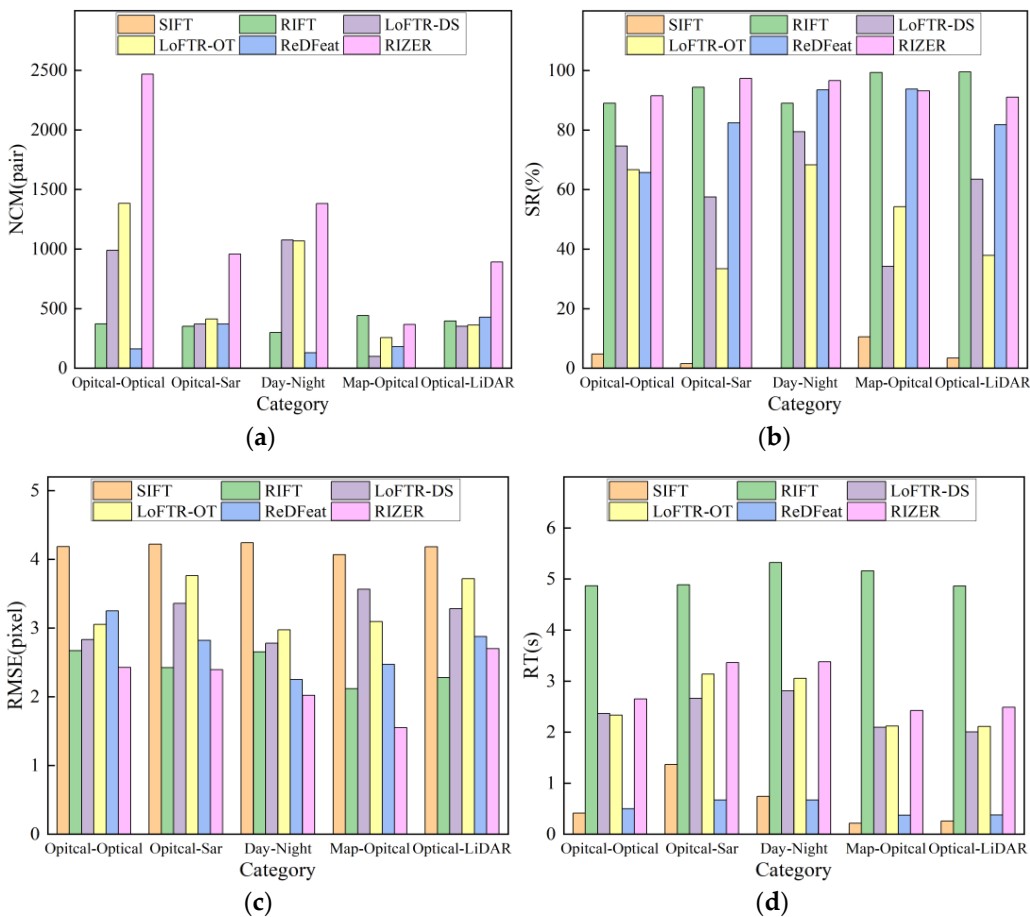

**Figure 9.** Indicator results of different algorithms for the registration of multimodal remote sensing images. (**a**) NCM; (**b**) SR; (**c**) RMSE; and (**d**) RT.

**Table 4.** Mean indicator values of the results of different algorithms for the registration of multimodal remote sensing images.

| Algorithm | NCM (Pair) | SR (%) | RMSE (Pixel) | RT (s) |
|---|---|---|---|---|
| SIFT | 1.60 | 4.04 | 4.18 | 0.60 |
| RIFT | 372.00 | 94.24 | 2.43 | 8.19 |
| LoFTR-DS | 577.20 | 61.86 | 3.16 | 2.39 |
| LoFTR-OT | 696.60 | 52.10 | 3.32 | 2.55 |
| ReDFeat | 253.60 | 83.44 | 2.73 | 0.52 |
| RIZER | 1213.20 | 93.92 | 2.22 | 2.86 |

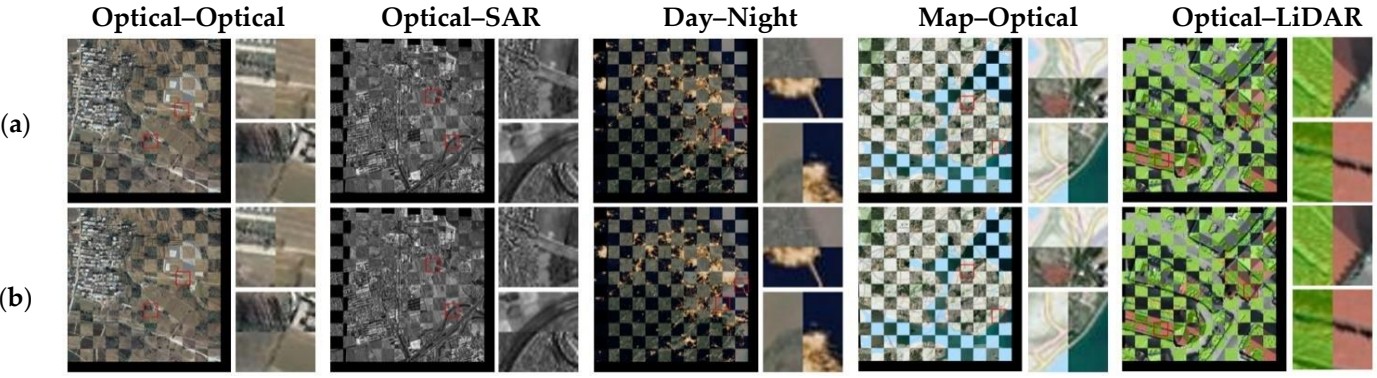

**Figure 10.** Registration checkerboard diagram of multimodal remote sensing images. (**a**) The real transformation; (**b**) RIZER.

*4.5. Ablation Study*

To validate the effectiveness of each proposed module, we conducted ablation experiments on the framework. Ablation experiments are a common technique in machine learning research that involve removing or substituting components of a model while keeping other factors unchanged. This allows assessing the individual contribution of each module to the overall performance. The four ablation experiments were designed as follows:

1. In order to verify the efficiency and accuracy of the HNSW algorithm [12] for finding the approximate nearest neighbors, we replaced it with the FLANN [51] algorithm and also used mutual matching for optimization; the rest of the algorithm remained unchanged.

2. In order to verify the validity of our proposed local geometric soft constraints module and the theory that a priori information about geometric structures has radiation variation invariance and is applicable to heterologous remote sensing image registration, we replaced it with the GMS algorithm [52] and likewise optimized the GMS algorithm by applying it to both the outlier removal of initial matches and incremental matching under multiple constraints.

3. In order to verify the validity of the least-squares fitting transform model that we used, and the proposed theory that coarse-level maps lead to an inaccurate estimation of the homography matrix and strong linear laws between matchpoints, we replaced the least squares fitting transformation model with the homography fitting transformation model, with its context remaining unchanged. We did not optimize the homography fitting algorithm and still adopted the RANSAC algorithm.

4. In order to verify the effectiveness as well as the efficiency of our proposed targeted outlier removal algorithm, we substituted the proposed approach with GC-RANSAC [53]. In addition, no optimization was performed on GC-RANSAC; instead, outliers were eliminated from all the matching points. Additionally, we devised an ablation study on the number of iterations to confirm that RIZER is capable of attaining optimal performance with an extremely low iteration count.

In order to fully demonstrate our incremental matching process and to better analyze the impact of the ablation experiments on each stage as well as on the final results, we recorded the necessary metrics for each stage, including the control point set $P_1^A$, the mutual three-nearest-neighbor matchpoint set $P_2^A$, the spatial structural similarity matchpoint set $P_3^A$, the coarse-level matchpoint set $P_c^A$, the fine-level matchpoint set $P_f^A$, and NCM, SR, RMSE, and RT. We randomly selected 50 pairs of images for testing from both the infrared and visible light test datasets, and the mean values of the quantitative evaluation indexes of the test results are shown in Table 5.

**Table 5.** Mean indicator values of ablation experiments.

| Metric | EXP 1 | EXP 2 | EXP 3 | EXP 4 | RIZER |
|---|---|---|---|---|---|
| $P_1^A(pair)\uparrow$ | 242.84 | 367.36 | 424.36 | 422.02 | 422.06 |
| $P_2^A(pair)\uparrow$ | 447.52 | 995.2 | 647.98 | 1068.74 | 1069.22 |
| $P_3^A(pair)\uparrow$ | 939.73 | 714.36 | 605.58 | 434.42 | 455.38 |
| $P_c^A(pair)\uparrow$ | 1630.09 | 2076.92 | 1677.92 | 1925.18 | 1946.66 |
| $P_f^A(pair)\uparrow$ | 1290.75 | 1373.96 | 891.32 | 1346.98 | 1500.58 |
| $NCM(pair)\uparrow$ | 1287.32 | 1363.68 | 889.66 | 1333.46 | 1496.72 |
| $SR(\%)\uparrow$ | 99.14% | 98.31% | 99.56% | 98.56% | 99.78% |
| $RMSE(pixel)\downarrow$ | 1.22 | 1.33 | 1.21 | 1.09 | 1.17 |
| $RT(s)\downarrow$ | 2.63 | 2.41 | 2.50 | 2.48 | 2.46 |

## 5. Disscussion

From Table 5, we can conclude that HNSW can find higher quality nearest neighbors by replacing the HNSW algorithm with the FLANN algorithm for the ablation experiment 1.

The hierarchical graph structure of the HNSW algorithm can approach the target nodes layer by layer to avoid finding only local optima. In contrast, the clustered kd-tree structure of FLANN is prone to falling into local optimal solutions. HNSW also utilizes the iterative adjustment process of inertial optimization to improve the accuracy of the indexing results further, so the HNSW algorithm is able to more fully utilize the complex feature vectors to obtain more accurate approximate K-nearest neighbor results. It is shown in Table 5 that the number of control points set $P_1^A$ obtained was higher, and it also had a direct impact on the subsequent mutual three-nearest-neighbor matchpoint set $P_2^A$. It can be seen that, by comparing the number of spatial structural similarities matchpoint set $P_3^A$, the result for ablation experiment 1 was more than that of RIZER, which indirectly reflects the touting of the incremental matching with the consistency of our local set structure, and even if feature vectors were not fully utilized above, it can be compensated for heavily at the end of the process, and ensure the optimal values of SR and RMSE. Moreover, the HNSW algorithm uses a graph-based incremental construction algorithm to approximate the target layer by layer, avoiding traversing all the data. In contrast, the FLANN algorithm adopts batch processing, which needs to traverse all the datasets to find the K-nearest neighbor during searching. Thus, whether it is the construction speed or the searching speed, the HNSW algorithm is faster. It reduced the overall running time by 0.17 s.

For ablation experiment 2, we replaced the outlier rejection algorithm with GMS. The core principle of GMS is that true correspondences often have more similar neighbors than false correspondences. From ablation experiment 2, it can be seen that GMS also has a certain invariance to radiation distortion but may remove some correct matches and some incorrect matches will also be included. This is reflected by the $P_1^A$ being lower, $P_3^A$ being higher, and $P_c^A$ being slightly higher compared to RIZER, and thus, the obtained $P_f^A$ is lower. The SR and RMSE performances were inferior to our local geometric consistency soft constraints, but the RT of the GMS algorithm was 0.05 s faster than RIZER. For ablation experiment 3, which used homography matrix fitting instead of least-squares fitting, the NC was reduced by 40.5%, as seen in Table 5, which directly verifies the regularity of the coarse-level graph and the effect on the homography. The use of homography matrix fitting drastically reduces the model's fitting accuracy, which manifests itself as a drastic reduction in the number of predicted correctly matched pairs of points, but due to the multiple constraints, it has a smaller impact on the overall SR and RMSE, further validating the value of multi-constrained incremental matching. If a homography matrix fitting algorithm with better performance is adopted or a more suitable fitting model is found, it may achieve a performance close to or exceeding that of RIZER.

For ablation study 4, it can be observed that our proposed targeted outlier removal algorithm achieved certain advantages over GC-RANSAC in terms of NCM and SR, with a 0.02 s faster RT, although with a slightly lower RMSE. Despite the average correct matching rate of 92.05% attained by RIZER without outlier removal, there exists some cases with an SR below 50%, posing a major challenge for RANSAC and its variants. Additionally, GC-RANSAC mainly tackles situations with numerous outliers, yet still produced an SR of 38.83% in the experiments, thus reducing the overall SR. We also evaluated the PROSAC algorithm [54], which is more heavily impacted by the presence of outliers, and obtained an SR of merely 96.99%. In contrast, our control point set registered an SR of 99.00%. Fitting homographies based on this facilitates convergence within fewer iterations whilst avoiding interference from copious outliers. As depicted in Table 6, we specified the maximum number of RANSAC iterations when using the control points to examine the proposed outlier removal technique. As expected, RANSAC ceased iterating upon convergence to the optimal result or reaching the ceiling on iterations. It can be discerned that given lesser iterations, RIZER exhibits slight errors in estimating homographies, principally with relatively lower NCM and SR values. However, within at most 10 iterations after convergence, RIZER attained the optimal registration accuracy. Thus, superior outlier removal efficiency was achieved. There are two possible optimization strategies for GC-RANSAC. One is to add GMS before it to initially eliminate outliers from all

matching points, which would improve the SR but decrease NCM. The other is to also use GC-RANSAC for homography matrix fitting based on the refined set of control points, and directly guide the elimination of other matching points. This may achieve performance equal to or slightly better than that of RIZER.

**Table 6.** Outlier removal iteration count test.

| Max Iteration Count | 1 | 3 | 5 | 10 | None |
|:---:|:---:|:---:|:---:|:---:|:---:|
| NCM (*pair*) ↑ | 1149.70 | 1367.96 | 1418.52 | 1482.18 | 1496.72 |
| SR (%) ↑ | 98.95 | 99.26 | 99.74 | 99.77 | 99.78 |

## 6. Conclusions

This paper proposed a new registration method named RIZER that can be applied to infrared and visible remote sensing images. Aimed at remote sensing scenes with sparse texture and the problem of less texture and fuzzy edges inherent in infrared images, we abandoned the use of multi-stage registration algorithms relying on feature-point detectors and instead innovatively adopted a detector-free, semi-dense, end-to-end matching algorithm which fundamentally solved the problem of the previous algorithms' dependence on corner points and edge points. To address the problem of the difficulty in obtaining heterogeneous remote sensing images with ground truth homography matrices for supervised domain-specific training, we adopted a pre-trained CNN combined with a Transformer structure to realize a data-driven approach of zero-shot learning, which provides a new paradigm in this field and establishes a test set of infrared and visible remote sensing images by combining generation with manual labeling. Then, a radiation-variation-invariant matching algorithm was proposed to address the problem of weak similarity of the corresponding positions due to the presence of radiation variations in heterologous sensors, which makes it difficult to establish effective matching. The graph model K-nearest neighbor algorithm of HNSW was pioneered in the field of image registration and used to establish mutual matching. The use of local geometric soft constraints independent of radiation variation was introduced. After that, the strategy of simulating the human eye for matching greatly reduced the difficulty of establishing matching for heterogeneous source images. Specifically, control point sets were established to simulate the salient regions easily found by the human eye after sweeping. The matching problem was transformed into a model fitting problem and the human eye was simulated to estimate the overall transformation pattern. An incremental matching method with multiple constraints was used, which could establish a numerical advantage in the matchpoints while guaranteeing a high accuracy rate. Finally, we proposed a targeted outlier rejection method, which enables the overall matching effect to realize further improvement with almost no increase in running time.

Compared with the LoFTR algorithm, RIZER improved the NCM by 25.9% at the expense of 16.4% of the RT. The RC reached 99.5% with an RMSE of 1.36 pixels, which is better than the state-of-the-art multimodal registration algorithms. RIZER also had good generalization ability in multimodal remote sensing scenarios. We experimentally demonstrated the feasibility of zero-shot learning for the Transformer module, the regularity between coarse-level matchpoints, and the geometric structure with radiation variation invariance. We also designed four ablation experiments to demonstrate the effectiveness of each module and analyzed the reasons in depth.

Of course, RIZER still has some limitations, mainly because it is overly dependent on the selection of control points. Hence, it is difficult to effectively obtain control points for some extreme cases, such as overlapping areas that are too small and rotation angles and scaling that are too large, which would result in a failure of the matching. Coarse-level matching is less effective for the registration of some low-resolution images. However, with the improvement of sensor pixels, RIZER will be more applicable, and most of the heterogeneous remote sensing registration scenarios have small parallax. More importantly,

the registration difficulties caused by radiation variation can be effectively solved by RIZER. The next step of our research will be to improve the Transformer module to improve the rotation and scaling performance and the radiation-variation-invariant module to further improve the registration effect for the above extreme scenarios by designing a loss function for training.

**Supplementary Materials:** The following supporting information can be downloaded at: https://www.mdpi.com/article/10.3390/rs16020214/s1. Figure S1. Matchpoint connectivity diagram using different algorithms for infrared and visible remote sensing images. (a) SIFT; (b) RIFT; (c) LoFTR-DS; (d) LoFTR-OT; (e) ReDFeat; and (f) RIZER. RIZER has significant advantages in terms of the number of correct matches. Overall, the matching points are evenly distributed across the entire overlapping region. RIZER adapts better to sparse texture regions and overcomes changes in natural scenes over time. RIZER also handles scale and rotation transformations well to some degree. Figure S2. Matchpoint connectivity diagram using different algorithms for multimodal remote sensing images. (a) SIFT; (b) RIFT; (c) LoFTR-DS; (d) LoFTR-OT; (e) ReDFeat; and (f) RIZER. RIZER achieved competitive matching performance across various scenes, but compared to matching between infrared and visible images, there are still some erroneous matches present. The datasets used in this paper are available in the following domain resources. Infrared and visible remote sensing image dataset: https://github.com/JasonLi-UCAS/NIR-VIS-RS.git (accessed on 4 January 2024); Utah AGRC aerial remote sensing dataset: https://downloads.greyc.fr/vedai/; Multimodal remote sensing image dataset: https://github.com/lan-cz/RISG-image-matching/tree/main/test.

**Author Contributions:** All authors were involved in the formulation of the problem and the design of the methodology; J.L. designed the experiment and wrote the manuscript; L.H. and G.B. reviewed and guided the paper; X.W. analyzed the accuracy of the experimental data; T.N. performed the data curation. All authors have read and agreed to the published version of the manuscript.

**Funding:** This work was supported by the National Natural Science Foundation of China (No. 62105328).

**Data Availability Statement:** The dataset and code will be released at https://github.com/JasonLi-UCAS/NIR-VIS-RS.git.

**Acknowledgments:** The authors thank the editors and reviewers for their hard work and valuable advice.

**Conflicts of Interest:** The authors declare no conflicts of interest.

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
