# Peer review of "Radiation-Variation Insensitive Coarse-to-Fine Image Registration for Infrared and Visible Remote Sensing Based on Zero-Shot Learning"

_remotesensing, doi:10.3390/rs16020214_

Round 1
Reviewer 1 Report
Comments and Suggestions for Authors
See attached file.

N/A
Reviewer 2 Report
Comments and Suggestions for Authors
The authors have developed a new algorithm for performing image registration between visible and infrared images collected via remote sensing. In general, the science is sound and the paper complete. For this reviewer a strength of the paper was that the data sets used for benchmarking cover a variety of objects/features as well as different modalities of remote sensing data which allows the robustness of the method to be assess. The largest weakness was the presentation can be hard to follow at times. Algorithmic ad other details are given without sufficient higher-level description. I recommend the following points be addressed prior to the article being published.
· Starting in the introduction the authors refer to “manual labeling” of images and lack of labeled image data. However, the work “label” can have multiple definitions in the context of images and it’s not clear what the authors are referring to. Do they mean “labeled” in terms of features (building/tree/river) or do they mean “labeled” in terms of labeled to indicate the points on one images that correspond to those on another (e.g. control points)? Please clarify and check the wording throughout.
· The authors refer to performing 4 “ablation experiments”. This review is unfamiliar with that term. These experiments consist of removing each module one at a time and substituting another state-of-the-art tool in its place to do the same task with everything else remaining the same. Is this a standard term for referring to these experiments? The term “substitution” of “validation” might be better if not. In the discussion surrounding these experiments please discuss the level of optimization performed with each of these cases and if not optimization was performed in the experiment, pleas comment on whether additional optimization could improve the methods that were compared relative to the already optimized method of the authors.
· Figure 1 contains a flow chart of the method. This chart is extremely hard to follow, even with the legend showing the different stages. I suggestion reorganizing it to contain the stages as blocks in a block diagram and using more space to achieve a more linear layout. Additionally, please label the starting point/raw data as well as the end,
· Suggest Figures 7 and 10 move to the supplementary information. The red and green lines to show correct and incorrect matches, cover the entire image scene, limiting their interpretation of the success to how much green and red there is visually. Figures 8 and 11 present the quantitative comparison which seems sufficient for this paper.
· It seems odd to call the work “our algorithm” throughout the entire paper. Recommend the authors select a name or acronym for this new method and define this at the beginning.
Comments on the Quality of English Language
In general the English language is acceptable, but it could be improved. The English language sentence structure is fine for short sentences, however for compound/longer sentences there's often too many clauses and sometimes the clauses are unrelated. Additionally, the vocabulary and word choice can be a confusing at times.
E.g. Page 2 "... which makes the data-driven no longer limited by domain specific data sets". Should this be "which makes the data-driven methods no longer limited by domain specific data sets".
Eg. Page 21: "After that, the strategy of simulating the human eye for matching by establishing the control point sets to simulate the salient regions found by the easy human eye after sweeping, by transforming the matching problem into a model fitting problem and simulating the human eye to estimate the overall transformation pattern, the difficulty of establishing the matching of heterogeneous source images is greatly reduced; an incremental matching method with multi-constrained is adopted, which can establish the numerical advantage of the match points while guaranteeing a high correctness rate." It is not possible to understand this sentence.
Example Page 21: "Finally, we propose a targeted outlier rejection method, which enables the overall matching effect to realize further improvement based on almost no increase in running time." should be ""Finally, we propose a targeted outlier rejection method, which enables the overall matching effect to realize further improvement with almost no increase in running time."
Round 2
Reviewer 1 Report
Comments and Suggestions for Authors
No further comments.
Reviewer 2 Report
Comments and Suggestions for Authors
The authors have done an excellent job of addressing my concerns and the paper is ready for publication at this time.